# Synergistic Larvicidal and Pupicidal Toxicity and the Morphological Impact of the Dengue Vector (*Aedes aegypti*) Induced by Geranial and *trans*-Cinnamaldehyde

**DOI:** 10.3390/insects15090714

**Published:** 2024-09-18

**Authors:** Sirawut Sittichok, Hataichanok Passara, Jirisuda Sinthusiri, Tanapoom Moungthipmalai, Cheepchanok Puwanard, Kouhei Murata, Mayura Soonwera

**Affiliations:** 1Office of Administrative Interdisciplinary Program on Agricultural Technology, School of Agricultural Technology, King Mongkut’s Institute of Technology Ladkrabang, Ladkrabang, Bangkok 10520, Thailand; best_pest22@hotmail.com (S.S.); hataichanok.pa@kmitl.ac.th (H.P.); 2Community Public Health Program, Faculty of Public and Environmental Health, Huachiew Chalermprakiet University, Bang Phli, Samut Prakan Province 10540, Thailand; jiri_ja@yahoo.com; 3Department of Plant Production Technology, School of Agricultural Technology, King Mongkut’s Institute of Technology Ladkrabang, Ladkrabang, Bangkok 10520, Thailand; 64604012@kmitl.ac.th (T.M.); 63604010@kmitl.ac.th (C.P.); 4School of Agriculture, Tokai University, Kumamoto 862-8652, Japan; kmurata@agri.u-tokai.ac.jp

**Keywords:** *Aedes aegypti*, geranial, *trans*-cinnamaldehyde, morphological changes, synergistic, non-target aquatic predator

## Abstract

**Simple Summary:**

Organic insecticides against *Aedes aegypti* L. mosquitoes are regarded as one of the most important tools for containing arboviruses transmitted by this species, such as dengue, Zika, and chikungunya in tropical countries including Thailand, considering the development of mosquito resistance to synthetic chemical larvicides. The most critical way to contain outbreaks is to control the vector that transmits pathogens. Monoterpenes are new, effective, and eco-friendly alternatives. This study investigated the larvicidal and pupicidal activities of two monoterpenes—geranial and *trans*-cinnamaldehyde—and their combinations. The efficacies of each formulation were compared against each other and temephos. Binary mixtures were significantly more effective than single formulations or temephos. They are safe for the non-target aquatic predator, guppies (*Poecilia reticulata*). These combinations make excellent insecticides for both larvae and pupae mosquito vector control, and the results open up the opportunity for both compounds to be developed into eco-friendly products.

**Abstract:**

Monoterpenes are effective and eco-friendly alternatives to conventional chemical larvicides. We tested single and binary mixtures of monoterpenes—geranial and *trans*-cinnamaldehyde—for their larvicidal and pupicidal activities against *Aedes aegypti* L. and for non-target toxicity on guppies (*Poecilia reticulata* Peters), using 1% (*w*/*w*) temephos as a reference. Geranial and *trans*-cinnamaldehyde at 250 ppm showed stronger larvicidal and pupicidal activities with a 100% mortality rate and an LT_50_ ranging from 0.3 to 0.6 h. All combinations were strongly synergistic against larvae and pupae compared to single formulations, with an increased mortality value (IMV) of 6% to 93%. The combination of geranial + *trans*-cinnamaldehyde (1:1) at 200 ppm showed the highest impact, with an IMV of 93%. The strongest larvicidal and pupicidal activities, a 100% mortality rate, and an LT_50_ of 0.2 h were achieved by geranial + *trans*-cinnamaldehyde (1:1) 500 ppm. They were thirty times more effective than a 1% temephos solution (LT_50_ ranging from 6.7 to 96 h) and caused obviously shriveled cuticles and a swollen respiratory system. All single and binary mixtures were not toxic to the guppies. Thus, the combination of geranial + *trans*-cinnamaldehyde has great potential as a safe insecticide for controlling mosquito larvae and pupae.

## 1. Introduction

The dengue (DENV), Zika (ZIKV), and chikungunya (CHIKV) viruses are diseases that share the same main vector—the *Aedes aegypti* L. (Diptera: Culicidae) mosquito [1]. Dengue is the most serious disease, leading to high morbidity and mortality in many parts of the world, including Thailand [2,3]. In 2023, Thailand presented the highest levels of DENV cases: ~17 thousand people were infected [3]. In the same vein, CHIKV spread nationwide and ~15 thousand confirmed cases were reported between 2018 and 2019 [4]. Regarding ZIKV disease, in 2019, a ZIKV outbreak in Thailand had a high impact on the public health system since it was closely related to newborn microcephaly and Guillain–Barré syndrome [5]. The World Health Organization (WHO) estimated that the annual economic loss due to mosquito-borne diseases was more than USD 10 billion [6], and ~USD 10 million in Thailand [7].

The main tactic for regulating mosquito populations is to deal with the insects while they are in the larvae and pupae stages [8]. Therefore, applying synthetic insecticides is the first and most common way to control mosquito larvae, pupae, or adults [9]. Temephos, a synthetic insecticide, is widely used for controlling larvae [10]. Unfortunately, temephos, apart from the high cost, was found to be highly toxic to the human nervous system and non-target species; further, mosquitoes have developed resistance to it [8,9,10]. Hence, one approach to this problem is to develop innovative insecticides based on essential oils (EOs), as secondary metabolites, which are characterized by strong bioactivity, because of synergism and multiple modes of action. Further, they are biodegradable and safe for humans and other aquatic species [8,11].

Monoterpenes were plant-produced secondary metabolites characterized by their volatility and fragrance. The monoterpenes represent about 90% of the essential oils and are the main ones responsible for insecticidal, larvicidal, and repellent activities [11,12]. Geranial, a monoterpenoid aldehyde, found abundantly in lemongrass (*Cymbopogon citratus*) [13], has shown insecticidal activity under laboratory conditions as a contact toxicant against *Ae*. *aegypti* eggs [8] and larvae [14], cabbage looper larvae (*Trichoplusia ni*) [15], and as a topical toxicant for housefly larvae and pupae (*Musca domestica*) [16] as a contact toxicant for housefly adults [17]. Trans-cinnamaldehyde, a monoterpenoid aldehyde extracted from cinnamon (*Cinnamomum verum*) [18], was found to be a contact toxicant for the larvae of *Anopheles gambiae* [12] and *Aedes albopictus* and *Culex quinquefasciatus* [19], and *Ae*. *aegypti* and *Ae*. *albopictus* adults [20], and had a strong repellent effect against adult female *Ae*. *aegypti* when sprayed [21].

In addition, several researchers reported that combinations of monoterpenes exhibited a higher degree of toxicity against mosquitoes and other insects than single monoterpene components (Table 1).

Furthermore, several combinations of monoterpenes showed high toxicity against several target species (*Ae*. *aegypti*, *Ae*. *Albopictus*, and *M*. *domestica*). Still, they were non-toxic to the tested other species (molly: *Poecilia latipinna*, guppy: *Poecilia reticulata*, zebrafish: *Danio rerio*, dwarf honeybee: *Apis florea*, yellow mealworm beetle: *Tenebrio molitor*, and a predatory bug: *Podisus nigrispinus*) [8,28,29,30,31]. Accordingly, investigations into insecticidal activities of geranial and *trans*-cinnamaldehyde against larvae and pupae of *Ae*. *aegypti* are required to better understand the complexity of their active mechanisms and enable effective management and control of these vectors.

Here, we investigated the insecticidal activity against *Ae*. *aegypti* larvae and pupae of several single and binary formulations of monoterpenes—geranial and *trans*-cinnamaldehyde. These two monoterpenes were chosen because they are generally known to be harmless for both humans and the environment with lower lethal dose (LD_50_) and lethal concentration (LC_50_) [32,33,34] and have insecticidal, antibacterial, and therapeutic properties [13,18]. The synergistic insecticidal effects and biosafety of the combinations against a common aquatic predator, guppy (*P*. *reticulata*), were evaluated. Guppy predators are widespread throughout Asia, including Thailand, and are an important part of the ecology [8,28]. The single and binary formulation morphological effects in mosquito larvae and pupae were observed using ultra-structural changes under scanning electron micrography (SEM). Reported insecticidal activity showed that they are a safe and sustainable alternative for managing mosquito populations in general and lowering breeding populations that merit more fieldwork [11,22,28].

## 2. Materials and Methods

### 2.1. Mosquito Breeding

*Ae*. *aegypti* mosquitoes were originally from the Department of Entomology, Armed Forces Research Institute of Medical Sciences (AFRIMS), Ratchatewi, Bangkok, Thailand. They were bred in the Laboratory of Entomology at the School of Agricultural Technology, King Mongkut’s Institute of Technology Ladkrabang (KMITL), Bangkok, Thailand, at 26.5 ± 2.0 °C, 75.5 ± 2.0% RH, and alternating 12 h light and dark periods. The hatched larvae were fed with special fish food pellets (Sakura^®^ gold, 35% high protein content: TSDP (Thailand) Co., Ltd., Samut Sakhon, Thailand). Larvicidal and pupicidal assays were used in the 4th larvae stage and 2-day-old pupae.

### 2.2. Chemicals

Geranial 96% (CAS-No: 5392-40-5, the major compound of lemongrass oil) and *trans*-cinnamaldehyde 98% (CAS-No: 104-55-2, the major compound of cinnamon oil) were purchased from Sigma-Aldrich Company Ltd. (Saint Louis, MO, USA). The 70% (*v*/*v*) stock solutions in ethanol were from T.S. Interlab Company Ltd., Bangkok, Thailand. Temephos (Sai GPO 1^®^, The Government Pharmaceutical Organization, Pathumthani, Thailand) was prepared at 1% (*w*/*w*), the concentration suggested by the World Health Organization for larval mosquitoes [35].

### 2.3. Treatment Formulations

The solvent for diluting the EO constituents was 70% (*v*/*v*) ethanol. The tested single formulations were geranial and *trans*-cinnamaldehyde at 50, 150, and 250 ppm each. The tested binary mixtures were *trans*-cinnamaldehyde + geranial (1:1) at 50, 200, 300, and 500 ppm. At these concentrations, the single compounds were shown to have insecticidal, ovicidal, and repellent effects against mosquitoes and houseflies [8,17,26,28]. The positive controls were 1% temephos at 1, 5, and 10 ppm.

### 2.4. Larvicidal and Pupicidal Assay

A dipping bioassay [36,37] was used to evaluate the larvicidal and pupicidal activities against *Ae*. *aegypti* under controlled laboratory conditions. Initially, the larvae were not fed with any nourishment. Ten 4th instar larvae or pupae of *Ae*. *aegypti* were put in a beaker containing each treatment: 1 mL of each essential oil formulation in 99 mL distilled water. Not being able to move to the surface and disturbing the water or not diving was the definitive indication that the larvae and pupae were dead. Larval mortality was monitored at 5, 15, and 30 min and 1, 6, and 24 h, while pupal mortality was monitored at 5, 15, and 30 min and 1, 6, 24, 48, and 72 h. The positive control, 1% temephos, was tested concurrently. Mortality rates of larvae and pupae (%M) were computed from the following [22]:Mortality rate (%M) = MT/TN × 100(1)
where MT is the total number of dead larvae or pupae mosquito; and TN is the total number of treated larvae or pupae mosquito.

Mortality index (MI) was determined from the following [37]:MI = %MT_treat_/%MT_temephos_(2)
where %MT_treat_ is the %mortality of the tested formulations at 24 h for larvae or at 72 h for pupae, and %MT_temephos_ is the %mortality of the 1% temephos at 10 ppm at 24 h for the larvae or 72 h for the pupae.

MI described the relative toxicity, with MI > 1 signifying that the treatment was more toxic to larvae or pupae than the temephos at 10 ppm in the control.

The increased mortality value (IMV) was the increased efficacy in larvicidal and pupicidal activities of binary mixtures versus single formulations. IMV was calculated from the following [20]:IMV = [%MT_mix_ − %MT_sing_/%MI_mix_] × 100(3)
where %MT_mix_ is the %mortality of the binary mixture at 24 h for larvae or 72 h for pupae, and %MT_sing_ is the %mortality of the single formulations at 24 for larvae or 72 h for pupae.

The synergistic mortality index (SMI) was the higher efficacy of binary mixtures over a single formulation at the same strength. SMI was calculated using the following [22]:SMI = LT_50 mix_/LT_50 sing_(4)
where LT_50 mix_ is the LT_50_ (50% lethal time) of the binary mixture and LT_50 sing_ is the LT_50_ of the single formulations.

Again, SMI indicates relative synergy, with SMI < 1 signifying that there is a synergistic effect, whereas SMI ≥ 1 indicates no synergy.

### 2.5. Microscopic and SEM Analysis of Morphological Changes

After the larvicidal and pupicidal bioassay, the morphological alterations and the external and internal changes in the treated larvae and pupae were observed by SEM at the Scientific and Technological Research Equipment, Chulalongkorn University, Pathumwan, Bangkok, Thailand [8].

SEM micrographs were captured at different magnifications in different body regions to view the external ultra-structural changes following treatment compared to untreated controls. After 24 or 72 h of treatment, samples were prepared in a fixative, 2.5% glutaraldehyde for 30 min in 0.1 M phosphate buffer, then thoroughly washed with the same buffer; then, the larvae and pupae were dehydrated by soaking in a series of ethanol solutions in water (30, 50, 70, and 95%). Each 1 h soaking with the ethanol solution was replicated three times with an automatic tissue processor. Then, the larvae and pupae were dried with a CO_2_ critical point drier. Each dehydrated sample was mounted on a stub coated with gold–palladium and examined by SEM (JEOL JSM-6610LV, Tokyo, Japan). 

### 2.6. Safety Bioassay of Non-Target Aquatic Predator

Guppy (*P*. *reticulata*) predators were purchased from an organic farm (Molly Fish Farm Thailand, Nakhon Chai Si District, Nakhon Pathom Province, Thailand (13.81556 °N, 100.03722 °E)). The toxicity of the tested formulations was tested against guppies, following Moungthipmalai et al. [8] and Selvi et al. [38]. One hundred fish were kept in a plastic container (400 × 600 × 300 mm) containing 60.0 L of clean water at 30 ± 2 °C and 74 ± 2% RH, with 12 h light and 12 h dark periods. The concentration of each treatment was 10,000 ppm following Soonwera et al. [28]. Ten adult male guppies were put in a plastic container (350 mm diameter, 180 mm height) containing 5 L of clean water. Each tested formulation was applied at 10,000 ppm. Temephos (1%) was tested concurrently. The mortality rate and abnormal behavior were recorded for 10 days post-treatment. Mortality rates (%MR) were computed from the following [37]:Mortality rate (%MR) = C/D × 100(5)
where C is the number of dead guppies and D is the number of treated guppies.

The biosafety index (BI) was determined using the following formula [28]:BI = %MT_non-target_/%MT_target_(6)
where %MT_non-target_ is the %mortality of non-target aquatic predators, and %MT_target_ is the %mortality of target species.

BI > 0.90 implies that the tested formulation was less safe than temephos for the aquatic predator, whereas BI < 0.90 implies that the formulation was safer than temephos.

### 2.7. Statistical Analysis

IBM’s SPSS Statistical Software Package version 28 (Armonk, NY, USA) was used for statistical analyses. All bioassays used a completely randomized design (CRD), and each treatment was replicated five times. The mean mortality rate for the larvicidal and pupicidal assay and the mean mortality rate for the non-target bioassay were analyzed by a one-way ANOVA, and Tukey’s test was used to investigate the differences across multiple treatment groups [39]. All statistical tests used the conventional *p* < 0.05 criterion. The time that a substance took to 50% mortality (LT_50_) against larvae and pupae was determined by probit analysis of mortality (number of larvae and pupae that had died at 24 and 72 h after exposure). Simple regression assessed the larvicidal and pupicidal efficacy against *Ae*. *aegypti* using generalized linear models (GLMs) with a binomial distribution [40]. The correlation coefficient, R^2^, was used to determine acceptable linearity.

## 3. Results

### 3.1. Larvicidal and Pupicidal Activities

Figure 1 shows regressions for larvicidal activity versus exposure time of single and binary mixtures against *Ae*. *aegypti*. All regression lines showed R^2^ close to 1. Several single formulations were less effective than any binary mixture after 24 h exposure. Among the single formulations, both geranial and *trans*-cinnamaldehyde at 250 ppm had more larvicidal activity (with a 100% mortality) than other formulations (with mortalities between 70% and 98%). All binary mixtures exhibited a strong larvicidal activity with 100% mortality, except geranial + *trans*-cinnamaldehyde (1:1) at 100 ppm (98% mortality). In contrast, 1% temephos had a mortality of only 96% at 1 ppm but 100% at 5 ppm or higher. In addition, for MI, both geranial and *trans*-cinnamaldehyde at 250 ppm and all binary mixtures showed equal effectiveness with 1% temephos 10 ppm, with MI = 1. Other formulations were less effective than 1% temephos 10 ppm with 0.7 ≤ MI ≤ 0.9.

Linear regression analyses of the pupicidal activity versus exposure time of single and binary mixtures against *Ae*. *aegypti* are shown in Figure 2. All obtained regression lines were positively linear with R^2^ close to 1. Several single formulations were less effective than all binary mixtures at 72 h after exposure. The highest mortality rate among the single formulations was 100%, achieved by geranial and *trans*-cinnamaldehyde at 250 ppm. The highest mortality for the binary mixtures was 100%, achieved by geranial + *trans*-cinnamaldehyde (1:1, 500 ppm). On the other hand, 1% temephos had a mortality of only 4% at 1 ppm rising to 90% at 5 ppm. For MI, the strongest activity was geranial 250 ppm, *trans*-cinnamaldehyde 250 ppm, and geranial + *trans*-cinnamaldehyde (1:1) 500 ppm. This MI was 1.1 times higher than 1% temephos (10 ppm). Other formulations were less effective than this temephos concentration with 0.1 ≤ MI ≤ 0.9.

Figure 3 shows the effect of larvicidal and pupicidal activities against *Ae*. *aegypti* in terms of LT_50_ value. All binary mixtures provided high larvicidal and pupicidal activities against *Ae*. *aegypti* than single formulations. Among single formulations, geranial 250 ppm showed the highest larvicidal and pupicidal activities against *Ae*. *aegypti* while *trans*-cinnamaldehyde 250 ppm had an LT_50_ of 0.6 h. In particular, the combination of geranial + *trans*-cinnamaldehyde (1:1) 500 ppm exhibited stronger larvicidal and pupicidal activities against *Ae*. *aegypti* with a short LT_50_ of 0.2 h which is more effective than 1% temephos 10 ppm (an LT_50_ of 0.5 h).

Moreover, the larvicidal and pupicidal efficacy of all binary mixtures against *Ae*. *aegypti* exhibited more than the efficacy of single formulations, with a synergistic mortality index (SMI) of 0.1–0.6. The highest synergy against both larvae and pupae was achieved by geranial + *trans*-cinnamaldehyde (1:1) 200 ppm, with an SMI of 0.1 and 0.2 (see Figure 4).

Fractions of IMV of single and binary mixtures against both larvae and pupae of *Ae*. *aegypti* are shown in Figure 5. Three mixtures improved by 6–93% compared to single formulations, except a combination of geranial + *trans*-cinnamaldehyde (1:1) 500 ppm showed no improvement. The highest increased mortality was achieved by geranial + *trans*-cinnamaldehyde (1:1, 200 ppm), with an IMV of between 22% and 93%.

### 3.2. Toxicity of Treatment on Morphological Changes

Larvae and pupae showed significant toxicity in response to varying treatment concentrations—see Figure 6 and Figure 7. Larvae and pupae treated with the binary mixture—geranial + *trans*-cinnamaldehyde—showed a remarkable reduction in the growth rate about twice that of the compounds alone. The control larvae showed a smooth cuticle, siphon, and intact anal papillae (Figure 6A–D). On the other hand, treated larvae showed highly damaged heads, abdomens, and thoraxes (Figure 6E,F) and swollen anal papillae (Figure 6G) as well as shrunken cuticles surrounding the respiratory siphon (Figure 6H). In particular, the respiratory siphon that has an anterolateral spiracular lobe, posterolateral spiracular lobe, lateral perispiracular lobe, anterior perispiracular lobe, and terminal spiracles were damaged, showing a conspicuous narrowing in the treated larvae compared to the control group.

In Figure 7A–D, it was evident that the pupae in the control group had smooth cuticles on the head, cephalothorax, and abdomen. However, the treated pupae showed significant cell damage in these areas (Figure 7E), along with swollen breathing openings and respiratory trumpets (Figure 7F), and shriveled cuticles around the genital lobe paddle (Figure 7H). Furthermore, the openings of the filter apparatus and pinna of the respiratory trumpets changed in the treated pupae (Figure 7G).

### 3.3. Efficacy of Non-Target Predators

The toxicity of single and binary mixtures on the mortality of adult guppies after 10 days of exposure is shown in Figure 8. All tested formulations showed low toxicity to adults with a mortality ranging from 2% to 12%. In contrast, 1% temephos 1 ppm was highly toxic to the adults, with a 100% mortality.

The BI is shown in Figure 9. All formulations provided a low BI, from 0.02 to 0.80 < 0.9, signifying that all treatments were extremely safe.

## 4. Discussion

Our previous and other studies reported good insecticidal and ovicidal activities against *Ae*. *aegypti* and *M*. *domestica* of single and binary mixtures of geranial and *trans*-cinnamaldehyde [8,16,17]. However, controlling the larvae and pupae should be the first line of defense against them [8]. Further, chemical insecticide use is associated with insecticide resistance, environmental damage, and public health risks [8,16,41]. Therefore, essential oils and their constituents represent an option for developing new agents due to their widely reported insecticidal activity [11,12,13,14,15]. It is significant that single and binary formulations of geranial and *trans*-cinnamaldehyde showed great potential as alternative agents against the early life stages of these pests.

In this study, all formulations showed stronger synergistic activities against larvae and pupae of *Ae*. *aegypti*, with shorter LT_50_ and higher mortality rate, IMV, and SMI, especially the combination of geranial + *trans*-cinnamaldehyde (1:1) 200 ppm, when compared with those of single formulation [16,42,43]. These results were consistent with other studies showing strong synergies, as well as our previous study, which reported that the combination of geranial + *trans*-cinnamaldehyde (1:1 at 10,000 ppm) showed a strong ovicidal activity against *Ae*. *aegypti* and *Ae*. *albopictus* [8]. Aungtikun et al. [17] found that a combination of (1:1) 1,8-cineole + geranial was strongly toxic to *Ae*. *aegypti* and *Ae*. *albopictus* adults. Soonwera et al. [26] concluded that a combination of geranial + *trans*-anethole was toxic to housefly adults. Similarly, benzyl benzoate + (E)-cinnamaldehyde (1:1) showed strong action against *Haemaphysalis longicornis* larvae [30], and Tak et al. [15] concluded that a 1:1 mixture of citral + limonene and citral + geranyl acetate exhibited strong activity against cabbage looper, *T*. *ni*.

Our MI showed that the combination of geranial + *trans*-cinnamaldehyde (1:1) 500 ppm was even more potent than temephos, a synthetic insecticide they intended to replace, and we concluded that *Ae*. *aegypti* larvae had developed resistance to temephos. We confirmed the findings of Soonwera et al. [37], who showed that 1% temephos was less effective in larvicidal and pupicidal activity than the tested essential oils. Similarly, Davila-Barboza et al. [44] reported that, in Mexico, *Ae*. *aegypti* larvae exhibited high resistance to temephos, with mortality ranging from 42% to 78% when exposed to 0.012 mg/L.

Micrographs showed extensive external morphological changes to larvae and pupae after treatment with all single and binary mixtures, similar to the work of Soonwera and Phasomkusolsil [43], who observed similar significant morphological changes and concluded that in larvae, after treatments with lemongrass oil, geranial was the major constituent. Also, Nakasen et al. [42] reported significant morphological larval changes in larvae *Ae*. *aegypti* induced by *C*. *verum* oil, of which *trans*-cinnamaldehyde was the major constituent. Gad [45] also observed major morphological changes in *Cx*. *quinquefasciatus* larvae after treatment with cinnamon oil (*C*. *osmophloeum*). Furthermore, Soonwera et al. [37] reported damage caused by d-limonene and *trans*-anethole and suggested their modes of action against *Ae*. *aegypti* were the same. Similar external changes were reported by Fujiwara et al. [46], caused by a combination of methyl cinnamate + linalool (1:4).

The insecticidal mechanism of action of geranial and *trans*-cinnamaldehyde demonstrated its interaction with acetylcholinesterase (AChE) and cytokinesis, indicating that monoterpene strongly interacted with both enzymes [20,22]. Both monoterpenes are toxic to *Ae*. *aegypti* larvae epithelial cells and respiratory siphons and pupae respiratory trumpet cuticles (see Figure 6 and Figure 7) [17,20]. Geranial is toxic to mosquitoes by inhibiting enzymes AChE. The acetylcholine esterase enzyme breaks down the neurotransmitter acetylcholine, which causes paralysis in the larvae and pupae and eventual mortality [17]. Trans-cinnamaldehyde caused mortality by inhibiting cytokinesis and reducing the ATPase activity of the cell membrane, causing decreased cell respiration and membrane integrity, and eventual mortality [20,47].

Monoterpenes are generally considered safe and eco-friendly for a wide range of other organisms, including aquatic predators, pollinators, and edible insects [8,28,29,30]. All tested formulations in this study were safe for adult guppies. Their BI was lower than 0.9 and the mortality was low. Moungthipmalai et al. [8] confirmed that geranial and *trans*-cinnamaldehyde and their combination (1:1) at 10,000 ppm showed very low guppy toxicity (LC_50_ = 37,000, 8700, and 4000 ppm) and molly (*P*. *latipinna*) (LC_50_ = 48,000, 8100, and 4300 ppm) whereas temephos 1 ppm showed 290 ppm for guppy and 500 ppm for molly. Soonwera et al. [22] reported that geranial and a combination of geranial + *trans*-anethole (1:1) were less toxic for stingless bees (*T*. *pegdeni*) (LT_50_ = 30.7 and 21.3 h). In addition, Nwanade et al. [30] indicated that *trans*-cinnamaldehyde at an LC_50_ of 28 µL/mL was not toxic to the yellow mealworm beetle (*T*. *molitor*) with 0% mortality. Similarly, Alsalhi et al. [48] reported that *trans*-cinnamaldehyde was not toxic to western mosquitofish, *Gambusia affinis* (LC_50_ = 3900 ppm), whereas temephos showed an LC_50_ = 4.8 ppm. On the other hand, 1% temephos was highly toxic to the guppy with a 100% mortality and BI = 1.0–1.09. Similarly, Moungthipmalai et al. [8] reported that temephos was highly toxic against guppy and molly with an LT_50_ of 300 and 500 ppm. Furthermore, temephos had highly dangerous neurotoxic side effects on humans with an acute oral LD_50_ > 2000 mg/kg for Wistar rats [10,49] and fish and aquatic organisms: western mosquitofish with an LC_50_ = 4.8 ppm, water beetle (*Acilius sulcatus* L.) with an LC_50_ = 1.1 ppm, and backswimmers (*Anisops bouvieri*) with an LC_50_ = 0.9 ppm [47]. In contrast, single and binary mixtures of geranial and *trans*-cinnamaldehyde were safe and benign to the tested aquatic predators. They were also safe and non-toxic to humans and other mammals at an LC_50_ < 100 ppm [32] with an oral LD_50_ > 2000 mg/kg for mice (geranial) [33] and a dermal LD_50_ > 1200 mg/kg for rats, an LD_50_ > 1000 mg/kg for rabbits [34], an oral LD_50_ > 2200 mg/kg for guinea pigs, and an LD_50_ >3000 mg/kg for rats (*trans*-cinnamaldehyde) [34], as well as were used as food additives in Asia since ancient times [22,34].

Therefore, the binary mixtures were extremely toxic to the larvae and pupae, causing rapid destruction and preventing any further growth.

## 5. Conclusions

Our findings suggest that the combination of geranial + *trans*-cinnamaldehyde (1:1) acted in synergy and was more effective than temephos. Thus, it was suitable for developing further into a larvicidal and pupicidal agent for managing mosquito populations. The mixture showed very promising larvicidal and pupicidal efficacy (at 100–200 ppm). Further, sublethal concentrations reduced the fertility of *Ae*. *aegypti*, and was safe for non-target organisms, especially guppies. In addition, bio-efficacy trials on this combination and toxicity investigations on human cells in lab and field study settings are potential future research topics. In addition, the combinations should be formulated as an aqueous solution for controlling mosquito immature stages (eggs, larvae, and pupae).

## Figures and Tables

**Figure 1 insects-15-00714-f001:**
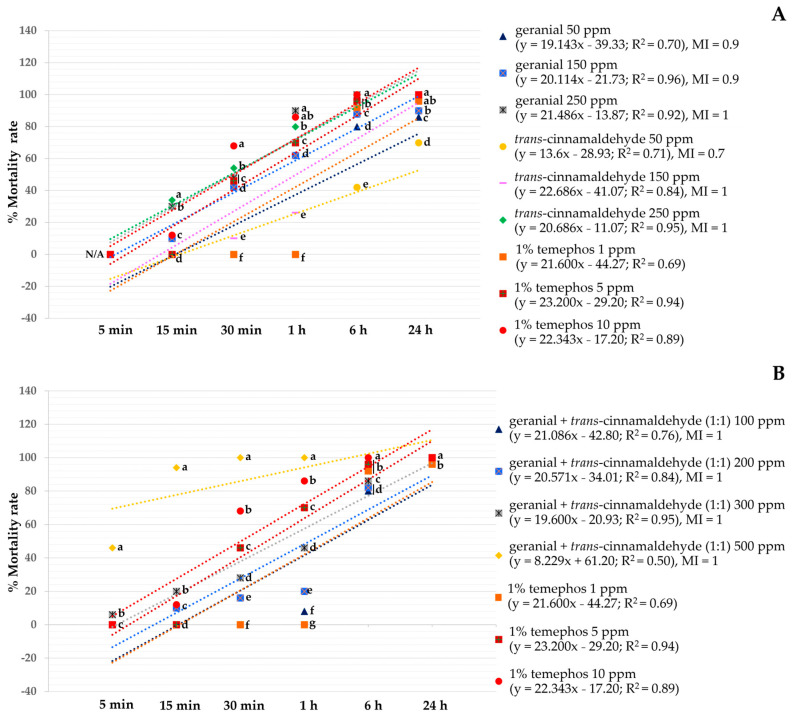
Mortality rate versus exposure time of test formulations against *Ae*. *aegypti* larvae: (**A**) single formulations and (**B**) binary mixtures. Note: The mean mortalities within a row (5, 15, and 30 min and 1, 6, and 24 h) labeled with different letters differ significantly by Tukey’s test.

**Figure 2 insects-15-00714-f002:**
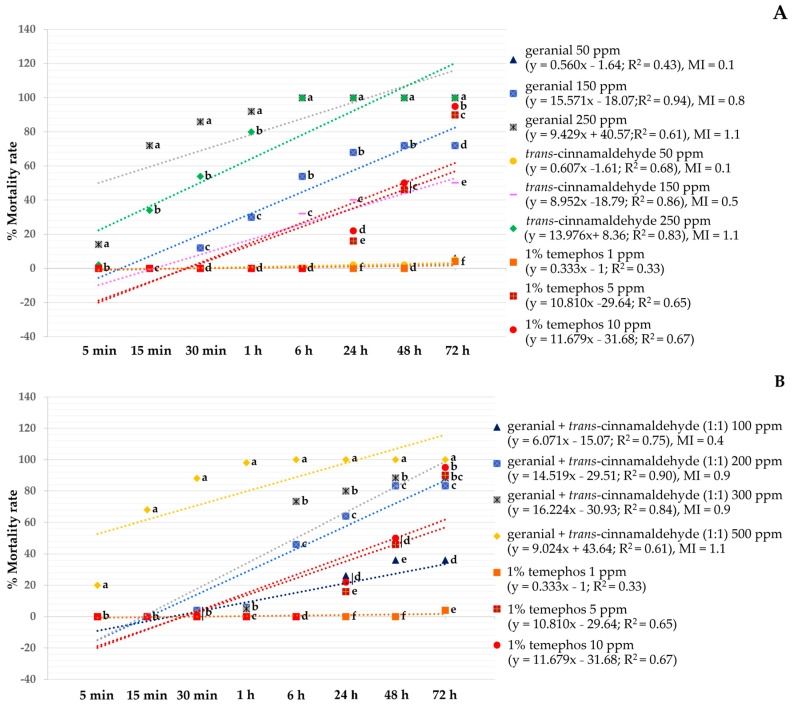
Mortality rate versus exposure time of test formulations against *Ae*. *aegypti* pupae: (**A**) single formulations and (**B**) binary mixtures.

**Figure 3 insects-15-00714-f003:**
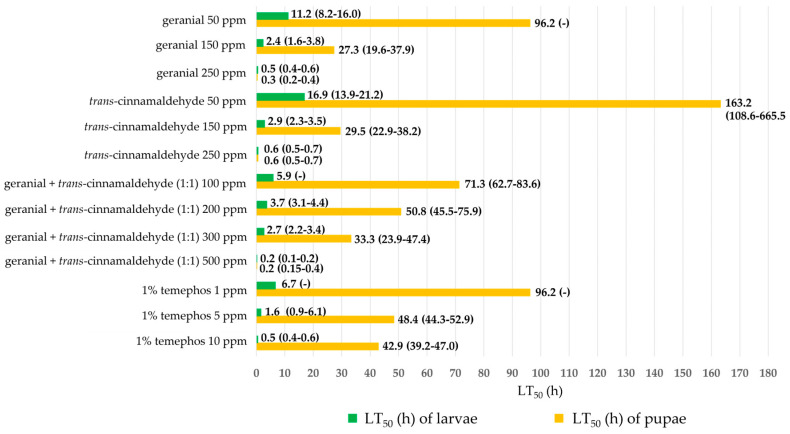
Lethal time 50% (LT_50_) of single and binary mixtures and 1% temephos against *Ae*. *aegypti* larvae and pupae. Note: LT_50_ = lethal time that kills 50% of the exposed organisms, *LL* = 95% lower confidence limit, and *UL* = 95% upper confidence limit.

**Figure 4 insects-15-00714-f004:**
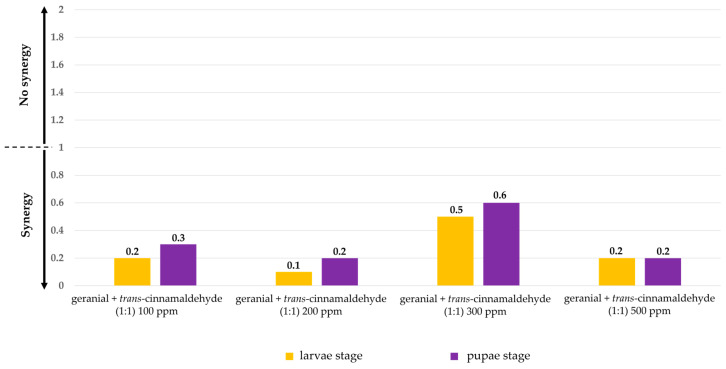
Synergistic mortality index (SMI) of several binary mixtures against larvae and pupae of *Ae*. *aegypti*.

**Figure 5 insects-15-00714-f005:**
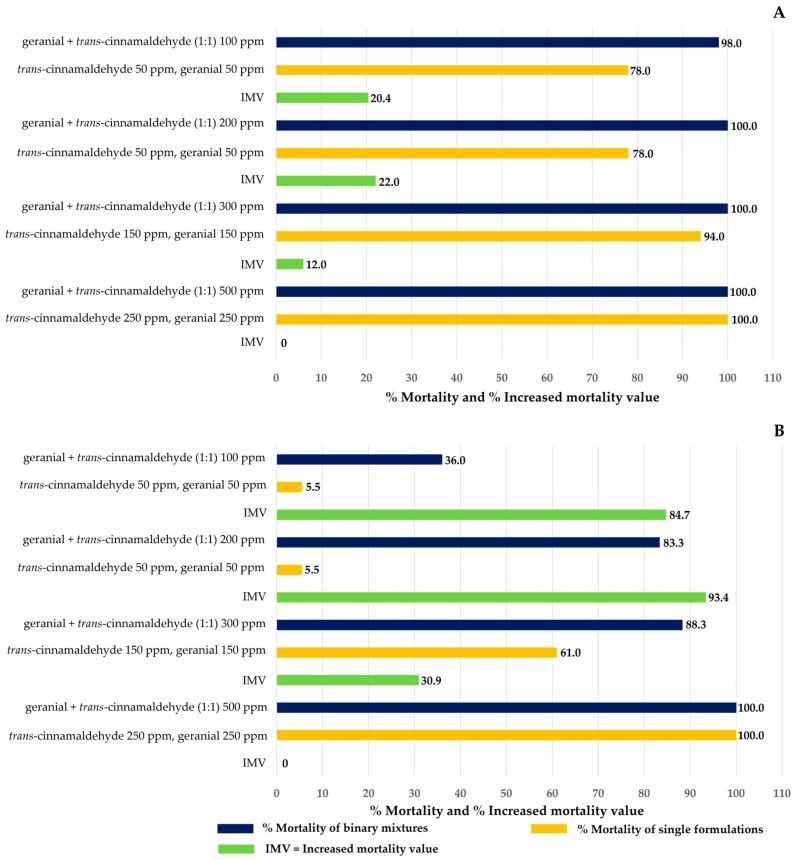
Increased mortality value (IMV) of binary mixtures against *Ae*. *aegypti* versus corresponding single formulations: (**A**) larvae and (**B**) pupae.

**Figure 6 insects-15-00714-f006:**
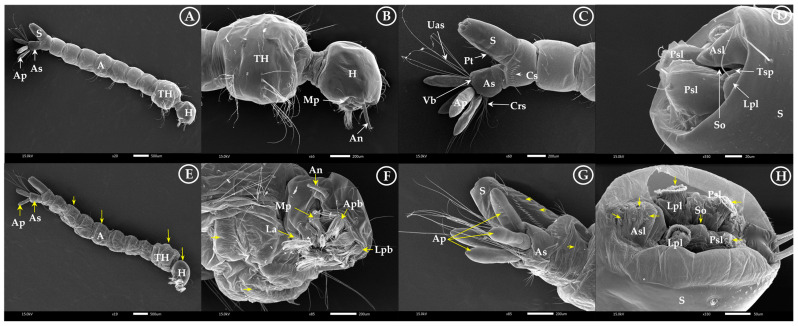
Scanning electron micrographs showing differences between untreated and treated larvae at the ultra-structural level: (**A**) normal larvae surface with head (H), thorax (TH), and abdomen (A); (**B**) normal lateral view of head capsule and mouth brushes of larvae with anteromedian palatal brush (Apb), lateral palatal brush (Lpb), antenna (An), maxillary palp (Mp), and labium (La); (**C**) normal final abdomen segments with abdominal segment (As), with comb scales (Cs), respiratory siphon (S) with pecten teeth (Pt), anal papillae (Ap), upper anal seta (Uas), cratal setae (Crs), and ventral brush (Vb); and (**D**) normal respiratory siphon with stigma opening (So), anterolateral spiracular lobe (Asl), posterolateral spiracular lobe (Psl), lateral perispiracular lobe (Lpl), anterior perispiracular lobe (Asl), and terminal spiracle (Tsp) (white arrow). Morphological changes showing damage and swelling of the anal papillae and respiratory siphon as well as abnormal head, abdomen, and thorax surfaces after the larvae were treated with single and binary mixtures of geranial and *trans*-cinnamaldehyde (**E**–**H**) (yellow arrow).

**Figure 7 insects-15-00714-f007:**
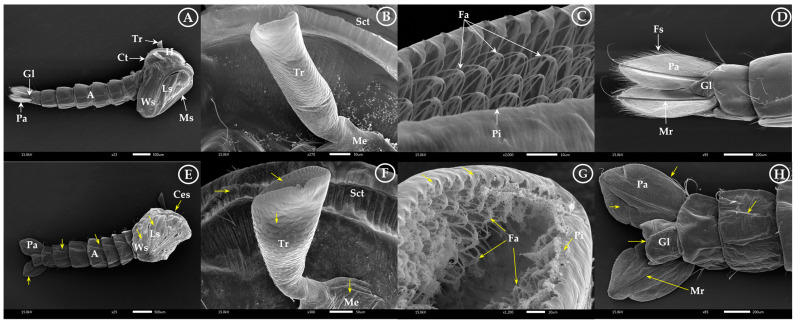
Morphological differences between untreated and treated pupae at the ultra-structural level: (**A**) normal pupae surface with head (H), cephalothorax (Ct), abdomen (A), compound eye sheath (Ces), trumpet (Tr), mouthpart sheath (Ms), leg sheath (Ls), and wing sheath (Ws); (**B**,**C**) normal lateral view of the respiratory trumpet and related structures with scutum (Sct), respiratory trumpet (Tr), filter apparatus (Fa), pinna (Pi), and meatus (Me); and (**D**) normal terminal abdominal structure with genital lobe (Gl), paddle (Pa), midrib (Mr), and filamentous spicule (fringes) (Fs) (white arrow). Morphological changes showing damage and swelling of the respiratory trumpet and related structures of the trumpet as well as an abnormality of the head, cephalothorax, abdomen, and terminal abdominal structure surfaces after the larvae were treated with geranial and *trans*-cinnamaldehyde (**E**–**H**) (yellow arrow).

**Figure 8 insects-15-00714-f008:**
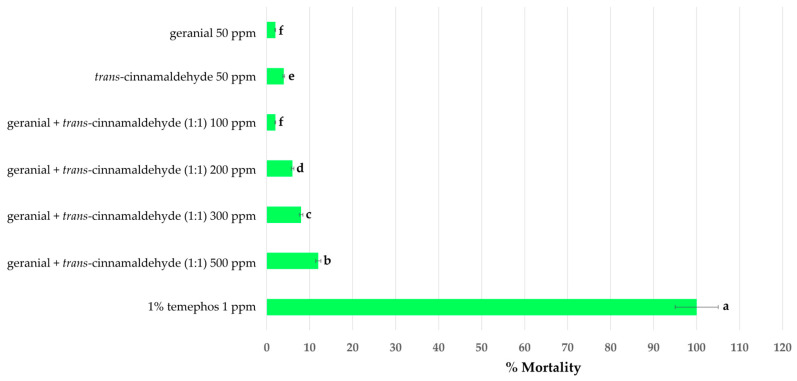
Mortality of single and binary mixtures and 1% (*w*/*w*) temephos 1 ppm in guppies at 10 days after treatment. Standard errors are indicated in the error bars. Note: mean mortalities within a row marked by a different letter differed significantly by Tukey’s test.

**Figure 9 insects-15-00714-f009:**
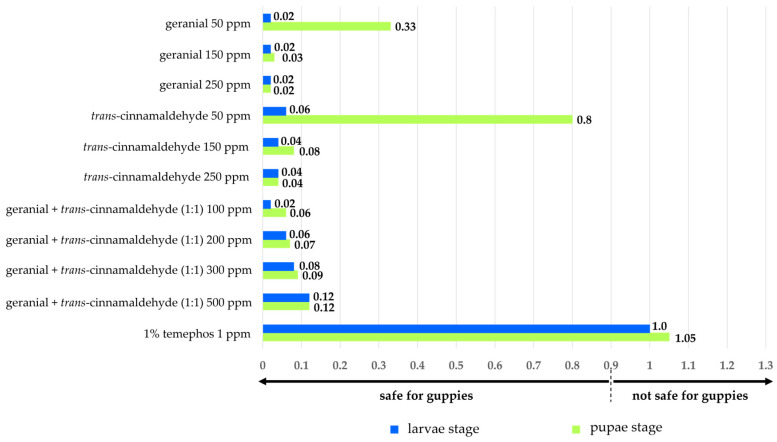
Biosafety index (BI) for guppies sharing an identical ecological niche as *Ae*. *aegypti* larvae and pupae, exposed to single and binary mixtures and 1% temephos 1 ppm.

**Table 1 insects-15-00714-t001:** Synergistic insecticidal combinations of monoterpenes against mosquitoes and houseflies.

Combination		Activity Against	Stage	References
d-limonene	geranial	*Ae*. *aegypti*	eggs	[8]
geranial	*trans*-cinnamaldehyde			
geranial	*trans*-anethole	*M*. *domestica*	adults	[17,22]
geranial	α-pinene			
*trans*-anethole	carvacrol	*Ae*. *aegypti*	larvae	[23]
R- (+)-limonene	*trans*-anethole			
eugenol	limonene	*Ae*. *aegypti*	larvae and adults	[24]
diallyldisulfide	limonene			
geraniol	citral	*Cx*. *quinquefasciatus*	larvae	[25]
1,8-cineole	geranial	*Ae*. *aegypti* and *Ae*. *albopictus*	adults	[26]
1,8-cineole	α-pinene	*Cx*. *pipiens*	adults	[27]
carvone	R (+)-pulegone			

## Data Availability

All relevant data are included in the article.

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
