# Peer review of "Synergistic Larvicidal and Pupicidal Toxicity and the Morphological Impact of the Dengue Vector (Aedes aegypti) Induced by Geranial and trans-Cinnamaldehyde"

_insects, 2024, doi:10.3390/insects15090714_

Round 1

Reviewer 1 Report

Comments and Suggestions for Authors

Thank you for arranging my review of the manuscript titled "Synergistic Larvicidal and Pulicidal Toxicity and Structural Changes of the Dengue Vector (Aedes aegypti) Induced by Geranial and trans-Cinnamaldehyde." This study investigates the efficacy and safety of essential oils at various concentrations, both individually and in combination, against the immature stages of Ae. aegypti, comparing their effects with temephos. Additionally, the authors assessed the morphological impacts of these essential oils using scanning electron microscopy.

However, all submitted images should be updated to higher resolutions and feature larger font sizes. Additionally, the authors need to include more detailed information in the Materials and Methods section, such as the susceptibility of the tested mosquitoes, criteria for selecting essential oils and their testing concentrations, and the conditions for collecting and observing post-exposure larvae and pupae. The discussion should also address the main gap between the mode of action of the essential oils and their morphological impact, incorporating relevant references.

Please refer to the specific comments provided below.

L2: Please consider replace ‘structural changes’ to ‘morphological impact’

L20: Need to be updated with recent updates with the dengue vaccine.

L25: Please add significantly

L28: please check the typo

L38: Revise it as 'They were thirty times more effective than a 1% temephos solution.'

L43: Please remove it.

Introduction

L72: laboratory

L73: cabbage looper larvae (Trichoplusia ni)

L74: please remove the Family Muscidae to keep manuscript in simple.

L87: M. domestica

L102: 'morphological effects' or 'morphological impact' might be clearer than 'mode of action' in the context of ultrastrucutral studies.

Materials and Methods

L109: Please add more details about the mosquitoes tested (e.g., susceptible or resistance?, obtained originally from?)

L111: 12 h

L112: please add manufacturer's location: Province, and Country.

L118: The 70% (v/v) stock solutions in ethanol were used as controls

L119: Is this the manufacturer? Please check it.

L121: remove Geneva, Switzerland

L122: Please add species name, amount, how did you keep the fish in your lab. etc.

L131: Please explain the reason why positive control tested in less concentration.

L143: MT, TN: stands for?

L167: Please include additional details such as the conditions for preserving the dead specimens and the time intervals for analysis (e.g., 1, 3, and 7 days post-treatment).

L175: 1 h

Results

Please re-submit all the figures in higher resolution and bigger font sizes.

L215: significantly more (P<0.05)?

L218: significantly lower mortality compared to the others?

L219: please add statistical information to compare the mortality between concentrations.

L225: The authors need to re-submit the images with higher resolution with larger font sizes (Figure 1)

L227: please check this again for the Figure.

L242: Please re-submit the images with higher resolution and larger font sizes (Figure 2)

L259: Please re-submit the images with higher resolution and larger font sizes (Figure 3)

L263: Re-submit the images (Figure 4)

L272: Re-submit the images (Figure 5)

L288: Re-submit the images (Figure 6)

L310: Re-submit the images (Figure 7). It may be worth to compare the morphological damages caused between temephos and EOs.

L327: Re-submit the images (Figure 8)

L332: Please mention that BI of 50 ppm trans-cinamaldehyde reached 0.8.

L329: please revise it.

Discussion

L361: Please revise this sentence with clear evidence from your findings and appropriate references to support. Please explain more about the meaning by 'a seriously harmful synthetic insecticide'. It means, the temephos killed all the tested non-target organisms?

L362: Have you compared mortality with susceptible strain of the Ae. aegypti? If so, please add resistant ratio.

L383: Is these morphological impacts linked with the enzymes? Please explain more about the ultrastructural damages caused by EOs.

L387: Please add more information related with the morphological damages.

L393: which means safer than temephos.

L394: please compare it with the concentrations of the positive control.

L397: compared to?

L399: zero mortality? please add indicated values.

L400: Please compare it with standardized positive control.

L401: Please add LC50.

L403: Revise it as LT requires time. Please add more references about temephos toxicity aginst fish and aquatic organisms.

L407: Convert it to ppm for comparison?

L408: Please standardize units to per kg.

L411: revise ‘, causing rapid destruction and preventing any further growth.’

Reviewer 2 Report

Comments and Suggestions for Authors

The authors studied the insecticidal activity against Ae. aegypti larvae and pupae of several single and binary formulations of monoterpenes—geranial and trans-cinnamaldehyde. At the same time, the synergistic insecticidal effects and biosafety of the combinations against a common aquatic predator, guppy (Poecilia reticulata), were tested.

The manuscript can be published, but only after the following modifications.

For the development of botanical larvicides, he generally considers EOs (monoterpenes) that show an LC50 of less than 100 ppm to be promising. However, at the end of the paper, the authors recommend a concentration of 500 ppm for the development of botanical larvicides. However, this is a concentration that is too high from an economic and application point of view. The authors should therefore better justify the results of their work and the final recommendations (or change them). It would therefore be appropriate to also discuss the influence of lethal and sublethal concentrations on the further development and fertility of mosquitoes (see e.g. DOI: 10.1016/j.indcrop.2021.114413) and insects in general (see e.g. DOI : 10.3390/plants13131863).

Non-standard ethanol was used as the carrier of the tested substances. But it is standard to use DMSO or Tween to emulsify. Justify the method used.

Reviewer 3 Report

Comments and Suggestions for Authors

1.       The text contains several instances of incorrect English usage that affect the clarity and overall meaning. I strongly recommend a thorough review of the manuscript for both grammatical errors and semantic accuracy.

Example Lines 29-31

The current sentence gives the impression that the larvicidal and pupicidal activities were tested on both Aedes aegypti (a mosquito species) and guppies (Poecilia reticulata, a non-target aquatic predator). This suggests that toxicity tests were conducted on both species. However, guppies are not relevant for larval and pupal stages; only adult guppies were used in the study to assess non-target toxicity. To avoid this confusion, I recommend revising the sentence to clarify that the larvicidal and pupicidal activities were tested exclusively on Aedes aegypti, while non-target toxicity was evaluated using adult guppies.

"Monoterpenes are effective and eco-friendly alternatives to conventional chemical larvicides. We tested single and binary mixtures of monoterpenes—geranial and trans-cinnamaldehyde—for their larvicidal and pupicidal activities against Aedes aegypti L. and for non-target toxicity on guppies (Poecilia reticulata Peters), using 1% (w/w) temephos as a reference."

The language used in the text should be more scientific. For example, in a sentence like 'All single and binary mixtures were not toxic to the fish,' the species name used in non-target toxicity tests should be specified instead of the general term 'fish.'

2.       All species and genus names should be written out in full and the describing author should be provided the first time they are mentioned; subsequently, abbreviations should be used throughout the text. For example, (Cymbopogon citratus, Musca domestica, Anopheles gambiae, Poecilia reticulata etc.)

3.       In the final part of the article, not all abbreviations used in the text have been included. If an abbreviation for any word or phrase is to be used in the text, it should be written out fully and clearly at its first occurrence, and then the abbreviation should be used throughout the text.

4. Lines 43-45: Keywords make more sense when listed in alphabetical order. In the keywords, 'L.' following the species name Aedes aegypti should be removed.

5.                      Lines 58-6 ; The first sentence states that the main tactic for regulating mosquito populations is to target the insects during their larval and pupal stages. However, the second sentence mentions the use of synthetic insecticides for both adults and larvae. This creates a contradiction with the earlier statement about focusing on early stages and disrupts the overall coherence of the text. A more suitable expression would focus specifically on control methods targeting only the larval and pupal stages.

6.       Lines 84-85; The title of Table 1, 'Synergistic insecticidal of monoterpenes combinations against mosquitoes and other insects,' is incorrect. The table only includes data on mosquitoes and houseflies, not other insects.

7.       In line 87, the phrase '(Ae. aegypti, Ae. albopictus, and housefly)' should be corrected by replacing 'housefly' with M. domestica.

8.       What is the age of the pupae used in the Materials and Methods section? For example, are they 1-2 days old? If so, it would be expected that they develop into adults within 2-3 days under the test conditions. This raises the question of how pupal mortality was assessed after 72 hours.

9.       The authors use the terms 'dose' and 'concentration' inconsistently throughout the text without considering their precise meanings. For example, should the term 'concentration' not be used when referring to '1% temephos'? Additionally, which term is considered when applying secondary components in an aquatic environment (EO constituents concentration)?

10.  The 'Predator Fish' section should be merged with the 'Safety Bioassay of Non-Target Aquatic Predator' section.

11.  Line 133 The authors mention that they performed ‘topical application’ on larvae and pupae in water, which contains a logical error. In the context of resistance tests, 'topical application' refers to the direct application of insecticides or other substances onto the surface of insects. This method is typically used to evaluate the effectiveness of insecticides against target pests by applying the substance directly to the insect's body. Applying a substance topically to larvae and pupae while they are submerged in water is inconsistent with the standard definition and use of the term 'topical application.

12.  It is stated that ten adult male guppies were used in the tests. What is the primary reason for using only male fish?

13.  The term '1% (w/w) temephos' should not include '(w/w)' after its first usage.

14.  The resolution of the graphs and figures is low; they need to be used in higher quality.

15.  The authors state, 'Our findings suggest that the combination of geranial + trans-cinnamaldehyde (1:1) 500 ppm should be developed further into a larvicidal and pupicidal agent for managing mosquito populations.' However, isn't 500 ppm a high concentration for a larvicide? Many larvicidal formulations are effective at much lower concentrations, often in the range of 1-100 ppm.

16.  Line 435- There are multiple individuals named "Prof. Dr. John Morris," each working in different fields and institutions.  Which university and department does he work in?

Comments on the Quality of English Language

1.       The text contains several instances of incorrect English usage that affect the clarity and overall meaning. I strongly recommend a thorough review of the manuscript for both grammatical errors and semantic accuracy.

Example Lines 29-31

The current sentence gives the impression that the larvicidal and pupicidal activities were tested on both Aedes aegypti (a mosquito species) and guppies (Poecilia reticulata, a non-target aquatic predator). This suggests that toxicity tests were conducted on both species. However, guppies are not relevant for larval and pupal stages; only adult guppies were used in the study to assess non-target toxicity. To avoid this confusion, I recommend revising the sentence to clarify that the larvicidal and pupicidal activities were tested exclusively on Aedes aegypti, while non-target toxicity was evaluated using adult guppies.

"Monoterpenes are effective and eco-friendly alternatives to conventional chemical larvicides. We tested single and binary mixtures of monoterpenes—geranial and trans-cinnamaldehyde—for their larvicidal and pupicidal activities against Aedes aegypti L. and for non-target toxicity on guppies (Poecilia reticulata Peters), using 1% (w/w) temephos as a reference."

The language used in the text should be more scientific. For example, in a sentence like 'All single and binary mixtures were not toxic to the fish,' the species name used in non-target toxicity tests should be specified instead of the general term 'fish.'

Round 2

Reviewer 1 Report

Comments and Suggestions for Authors

The authors have markedly improved their manuscript. I am pleased to approve it for publication.

Reviewer 2 Report

Comments and Suggestions for Authors

The authors supplemented and corrected the manuscript according to the comments of the reviewers, I have no further comments.

Reviewer 3 Report

Comments and Suggestions for Authors

All species names in the text should be italicized. Still some species names are not italicized

Remove author names after species (L.) names in the table

In the toxicity tests on pupae, it is said that 2-day-old pupae were used, but at 26 degrees Celsius, the pupa becomes adult at the end of 3 days during the test period. I wonder if they used pupae at an earlier age in toxicity tests.
